# Positive Fecal Immunochemical Test Results Are Associated with Increased Risks of Esophageal, Stomach, and Small Intestine Cancers

**DOI:** 10.3390/jcm9072172

**Published:** 2020-07-09

**Authors:** Yoon Suk Jung, Jinhee Lee, Chang Mo Moon

**Affiliations:** 1Division of Gastroenterology, Department of Internal Medicine, Kangbuk Samsung Hospital, Sungkyunkwan University School of Medicine, Seoul 03181, Korea; ys810.jung@samsung.com; 2Department of Endocrinology and Metabolism, Ajou University School of Medicine, Suwon 16499, Korea; marie0715@naver.com; 3Department of Internal Medicine, College of Medicine, Ewha Womans University, Seoul 07985, Korea

**Keywords:** fecal immunochemical test, esophageal cancer, gastric cancer, small intestine cancer

## Abstract

**Background:** The current guideline does not recommend upper gastrointestinal evaluation for patients with a positive fecal immunochemical test (FIT) and negative colonoscopy results. However, this indication was based on low-quality evidence as data on this issue are very limited. We assessed the risk of proximal cancers (oral or throat, esophageal, stomach, and small intestine cancers) after negative or positive FIT results in the Korean National Cancer Screening Program (NCSP). **Methods:** Using the NCSP databases, we collected data on participants who underwent FIT between 2009 and 2011. Participants were classified based on FIT results and colorectal cancer (CRC) diagnosed within 1 year after FIT as FIT− (*n* = 5,551,755), FIT+/CRC− (*n* = 368,553), and FIT+/CRC+ (*n* = 12,236). **Results:** The incidence rates of overall proximal cancers in FIT−, FIT+/CRC−, and FIT+/CRC+ patients within 1, 2, and 3 years after FIT were 0.38%, 0.68%, and 2.26%; 0.57%, 0.93%, and 2.74%; and 0.79%, 1.21%, and 3.15%, respectively. After adjusting confounding variables, the risks of esophageal, stomach, and small intestine cancers as well as overall proximal cancers within 1, 2, and 3 years after FIT were higher in FIT+/CRC− patients than those in FIT− patients. However, the risk of oral or throat cancer did not differ between FIT− and FIT+/CRC− patients. The risks for oral or throat cancer and small intestine cancer were higher in FIT+/CRC+ patients than those in FIT+/CRC− patients. **Conclusions:** In this population-based study, FIT+/CRC− patients were at higher risk for esophageal, stomach, and small intestine cancers than were FIT− patients, suggesting that positive FIT results were associated with these cancers.

## 1. Introduction

The fecal immunochemical test (FIT) has been widely recommended by guidelines for colorectal cancer (CRC) screening, based on the results of randomized controlled trials showing reductions in CRC incidence and mortality [1,2]. FITs directly measure human hemoglobin in stool using specific monoclonal or polyclonal antibodies that selectively react with the globin portion of human hemoglobin [3,4]. Since hemoglobin is degraded or diluted as it moves through the gastrointestinal (GI) tract, FITs are considered specific for lower GI tract bleeding [5]. Therefore, FIT findings are less likely to be positive in patients with upper GI (UGI) cancer. However, clinical data on this issue are extremely limited.

The U.S. Multi-Society Task Force on CRC suggests that in the absence of symptoms or signs of upper GI disease, positive FIT and negative colonoscopy results should not prompt UGI assessment [4]. However, this is a weak recommendation with very low-quality evidence. Although previous studies evaluated UGI outcomes in patients who were fecal occult blood test (FOBT)-positive and colonoscopy-negative, their results were inconsistent and the numbers of patients included were too small to reach a firm conclusion [6,7,8,9,10,11,12]. Furthermore, previous studies were mainly based on guaiac FOBT (gFOBT), which is not specific for human blood and has a poorer sensitivity for CRC detection than that of the FIT [6,13]. It remains uncertain whether patients with positive FIT and negative colonoscopy results are more likely to have cancers in the UGI tract compared to patients with negative FIT results. Large-scale research is required to clarify this issue.

Therefore, we conducted a nationwide population-based study to assess the risk of developing proximal cancers (oral or throat, esophageal, stomach, or small intestine cancers) within 1, 2, and 3 years after a positive or negative FIT result in an annual FIT-based CRC screening program. We compared the risks of proximal cancers between three groups classified on the basis of FIT results and CRC status (FIT−, FIT+/CRC−, and FIT+/CRC+).

## 2. Materials and Methods

### 2.1. Study Population

The Korean government supports CRC screening via the National Cancer Screening Program (NCSP). The NCSP provides a single annual FIT for all Koreans aged 50 years or older as an initial CRC screening and a colonoscopy as a second test for those with positive FIT findings. Data were extracted from the National Health Information Database (NHID) of the National Health Insurance Service (NHIS) running the NCSP. The study population comprised participants who underwent FITs through the NCSP between 1 January 2009, and 31 December 2011. We only collected the initial FIT results for those who underwent two or more FITs during the study period.

Of the 6,343,048 participants selected, we excluded those with previous diagnoses of cancer (including CRC) (*n* = 370,340) or inflammatory bowel disease (*n* = 22,073). Another 18,091 participants were excluded due to missing data on screening date, age, and sex. Finally, this study included a total of 5,932,544 participants (Figure 1).

The NHIS-NHID is encrypted and does not contain personal identifiers. This study protocol was approved by the Institutional Review Board (IRB) of Ewha Womans University Mokdong Hospital (IRB No. 2020-02-029).

### 2.2. Definition of Variables and Ascertainment of Cancers

The NHIS-NHID database contains information on comorbidities (e.g., cancer) based on International Classification of Disease 10th revision (ICD-10) codes as well as age, sex, screening date, and FIT results (negative, positive) for all participants. Information regarding various clinical factors such as health-related behavior, body mass index (BMI) and medication use was also obtained from the NHIS-NHID. Smoking, drinking habits, and a family history of any cancer were assessed from data of the medical questionnaires in the NHIS-NHID. In this study, current smoking and alcohol drinking more than once a week were included as covariates in the multivariable analyses. Diabetes mellitus (DM) was defined as having the diagnostic code (E11–E14) prior to FIT. Use of aspirin was defined as the total prescription days of aspirin more than 180 days during 2 years prior to FIT.

The participants were classified based on FIT results and CRC (ICD-10: C18~C21, D01.0~D01.3) diagnosed within 1 year after FIT as follows: Group 1, FIT-negative participants (FIT−); Group 2, FIT-positive participants who were not diagnosed with CRC within 1 year after FIT (FIT+/CRC−); and Group 3, FIT-positive participants who were diagnosed with CRC within 1 year after FIT (FIT+/CRC+).

Proximal cancers were defined as oral or throat (C00–14), esophageal (C15), stomach (C16), and small intestine cancers (C17). Oral or throat locations included the lip, tongue, gum, mouth, palate, major salivary glands, parotid gland, oropharynx, nasopharynx, tonsil, piriform sinus, and hypopharynx. We also compared the risks of hepatopancreatobiliary cancers (C22–25) among the three groups. The hepatopancreatobiliary locations included the liver, gallbladder, biliary tract, and pancreas.

To increase definition accuracies, we defined cancers as having both the cancer registration code and the appropriate diagnostic code. The Korean government (NHIS) manages a registration program for all cancers to subsidize the medical expenses of patients with cancer. Through this program, the Korean government covers 95% of hospital expenses related to cancer up to 5 years after diagnosis; thus, patients pay only 5% of their hospital expenses during this period. Upon cancer diagnosis, the doctor registers the patient in the program and the patient is assigned an exempted calculation code. A cancer diagnosis is unlikely to be missed to ensure access to these medical benefits. Conversely, cancer diagnoses should be confirmed using strict criteria based on histological examination for registration in the program. The diagnosis date of cancers was defined as the date when both diagnostic and exempted calculation codes for cancer were registered in the NHIS-NHID database.

### 2.3. Statistical Analysis

Baseline characteristics were compared among the three groups by $\chi^{2}$ tests and one-way analysis of variance for categorical and continuous variables, respectively. We performed logistic regression analysis to estimate the odds ratios (ORs) with 95% confidence intervals (CIs) of the risk of developing proximal and hepatopancreatobiliary cancers within 1, 2, and 3 years after FIT among the three groups. The results were presented as multivariable-adjusted and crude ORs. We adjusted potentially confounding variables including age, sex, current smoking, alcohol intake, obesity (BMI ≥ 25 kg/m^2^), DM, family history of any cancer, use of aspirin, and esophagogastroduodenoscopy (EGD) within 1 year after FIT.

All reported *p*-values were two-tailed, and *p* < 0.05 was considered statistically significant. All data analyses were performed using SAS software version 9.4 (SAS Institute, Cary, NC, USA).

## 3. Results

### 3.1. Baseline Characteristics of the Study Population

The study population was divided into three groups as follows: Group 1 (FIT−), *n* = 5,551,755; Group 2 (FIT+/CRC−), *n* = 368,553; and Group 3 (FIT+/CRC+), *n* = 12,236 (Figure 1). The baseline characteristics of these groups are shown in Table 1. The mean age of the study population was 60.6 ± 8.2 years and 44.0% were men. The mean ages and proportions of men were highest in Group 3 and lowest in Group 1. The proportions of current smoking, alcohol intake, DM, and aspirin use and the proportion of patients who underwent EGD within 1 year after FIT were also highest in Group 3 and lowest in Group 1. The proportion of obese patients was highest in Group 2, while the proportion of patients with a family history of any cancer was highest in Group 1.

### 3.2. Risk of Proximal Cancers According to FIT Results and Presence of CRC

The incidences of cancers diagnosed per location for each group are shown in Table 2 and the risks of proximal cancers between the three groups are compared in Table 3. The incidence rates of overall proximal cancers in Groups 1, 2, and 3 were 0.38%, 0.68%, and 2.26% within 1 year after FIT; 0.57%, 0.93%, and 2.74% within 2 years after FIT; and 0.79%, 1.21%, and 3.15% within 3 years after FIT, respectively. Compared with those of Group 1, the multivariable-adjusted ORs for overall proximal cancers within 1, 2, and 3 years after FIT were higher in Group 2 (1.47 [95% CI, 1.41–1.54], 1.38 [95% CI, 1.33–1.43], 1.31 [95% CI, 1.27–1.36], respectively) and Group 3 (1.50 [95% CI, 1.32–1.69], 1.51 [95% CI, 1.35–1.69], 1.48 [95% CI, 1.33–1.64], respectively).

The incidence rates of oral or throat cancer in Groups 1, 2, and 3 were 0.01%, 0.02%, and 0.10% within 1 year after FIT; 0.03%, 0.04%, and 0.11% within 2 years after FIT; and 0.05%, 0.06%, and 0.15% within 3 years after FIT, respectively (Table 2). The adjusted ORs for oral or throat cancer within 1, 2, and 3 years after FIT were higher in Group 3 than those in Group 1 but did not differ significantly between Groups 1 and 2 (Table 3).

The incidence rates of esophageal cancer in Groups 1, 2, and 3 were 0.02%, 0.04%, and 0.08% within 1 year after FIT; 0.03%, 0.06%, and 0.15% within 2 years after FIT; and 0.05%, 0.08%, and 0.20% within 3 years after FIT, respectively (Table 2). The adjusted ORs for esophageal cancer within 1, 2, and 3 years after FIT were higher in Group 2 than those in Group 1 (1.36 [95% CI, 1.14–1.63], 1.32 [95% CI, 1.14–1.52], 1.29 [95% CI, 1.14–1.46]) but did not differ significantly between Groups 1 and 3 (Table 3).

The incidence rates of stomach cancer in Groups 1, 2, and 3 were 0.35%, 0.63%, and 1.90% within 1 year after FIT; 0.50%, 0.83%, and 2.30% within 2 years after FIT; and 0.69%, 1.07%, and 2.62% within 3 years after FIT, respectively. The incidence rates of small intestine cancer in Groups 1, 2, and 3 were 0.005%, 0.01%, and 0.24% within 1 year after FIT; 0.01%, 0.02%, and 0.27% within 2 years after FIT; and 0.01%, 0.02%, and 0.31% within 3 years after FIT, respectively. Lastly, the incidence rates of hepatopancreatobiliary cancer in Groups 1, 2, and 3 were 0.17%, 0.23%, and 2.11% within 1 year after FIT; 0.30%, 0.41%, and 2.86% within 2 years after FIT; and 0.45%, 0.60%, and 3.54% within 3 years after FIT, respectively (Table 2). Compared to those in Group 1, the risks for stomach, small intestine, and hepatopancreatobiliary cancers within 1 year after FIT were higher in Group 2 (adjusted OR [95% CI]; 1.49 [1.42–1.55], 1.74 [1.24–2.45], and 1.20 [1.12–1.28], respectively) and Group 3 (adjusted OR [95% CI]; 1.36 [1.19–1.55], 14.39 [9.75–21.25], and 4.86 [4.27–5.52], respectively). The risks for these cancers within 2 and 3 years after FIT were also higher in Groups 2 and 3 than those in Group 1 (Table 3).

Table 4 shows comparisons of the risks of proximal cancers between Groups 2 and 3. The risk of overall proximal cancers within 3 years after FIT and the risks of oral or throat cancer within 1 and 3 years after FIT were higher in Group 3 than those in Group 2. The risks of small intestine cancer and hepatopancreatobiliary cancer within 1, 2, and 3 years after FIT were also higher in Group 3 than those in Group 2. However, the risks of esophageal and stomach cancer did not differ significantly between Groups 2 and 3.

## 4. Discussion

The results of this Korean population-based study showed that FIT+/CRC− patients were at a higher risk of UGI cancers than FIT− patients. More specifically, the risks of esophageal, stomach, and small intestinal cancers in FIT+/CRC− patients were higher than those in FIT− patients.

Previous studies reported very low frequencies of esophageal/gastric cancer in FOBT-positive and colonoscopy-negative patients. Three studies (*n* = 70, 211, and 70, respectively) found no cases of esophageal/gastric cancer [7,8,9]. Two studies reported esophageal/gastric cancer rates of approximately 1% (*n* = 5/498 and *n* = 1/74, respectively) [10,11], while one study reported a gastric cancer rate of 7%, corresponding to one case out of 14 patients with UGI symptoms [12]. However, these studies only reported the frequencies of UGI cancer in patients with positive FOBT and negative colonoscopy results and did not compare these rates with those in FIT-negative patients. Moreover, these studies used gFOBT, which is less accurate for detecting CRC than FIT, and the sample sizes were too small for statistical confidence in the findings.

To date, few studies have compared the risk of gastric or esophageal cancer according to FIT and/or colonoscopy results. Contrary to our results, a Japanese study showed no difference in the detection rates of gastric cancer between FIT− and FIT+ screenees (0.13% [*n* = 22/17,352] vs. 0.15% [*n* = 2/1372]) [14]. In this study, however, radiological imaging testing by barium meal rather than EGD was used to detect gastric cancer [14]. Thus, the accuracy of the gastric cancer detection rate may have been lower. A Danish study including 20,671 screenees reported a significantly higher incidence of gastric and esophageal cancers within 2 years after gFOBT in gFOBT+ persons than that in gFOBT− individuals [15]. However, the study concluded that, based on the low positive predictive value (PPV) of gFOBT (0.52%), UGI examination was not justified in asymptomatic gFOBT+ and CRC− individuals [15]. An Italian study assessing gastric cancer incidence in FOBT− (*n* = 83,489), FOBT+/colonoscopy− (*n* = 3555), and FOBT+/colonoscopy+ (*n* = 2025) subjects compared with expected standardized incidence rates revealed that the risk of developing gastric cancer within 1 year after FOBT+/colonoscopy− was approximately four times higher than that in the general population, while the risks within 3 years after FOBT− and FOBT+/colonoscopy+ findings were not significantly different compared with that in the general population [16]. Although this study showed an increased short-term risk of gastric cancer in FOBT+/colonoscopy− subjects, the authors also concluded that recommending routine EGD for these subjects was questionable because of the low PPV of FOBT (0.4% [*n* = 14/3555]) [16].

Similar to our study, a recent Dutch study compared the risk of proximal GI cancers (oral cavity, throat, esophageal, stomach, and small intestine cancers) diagnosed within 3 years after FIT among three groups classified based on FIT results and colonoscopy findings [17]. However, this study showed different results than ours. The aforementioned study observed no significant differences between FIT− (0.31% [*n* = 44/14001]), FIT+/colonoscopy− (0.44% [*n* = 6/1365]), and FIT+/colonoscopy+ subjects (0.28% [*n* = 2/722]) (*p* = 0.72) [17].

This is the first study to show increased short- and long-term risks for esophageal, stomach, and small intestine cancers in FIT+/CRC− patients than those in FIT− patients through an analysis of a nationwide population-based database. Our study, which included a relatively very large sample size (*n* = 5,932,544), provides reliable information on this topic. In particular, since esophageal and small intestine cancers are not common malignancies, it may be difficult to compare the incidences of these cancers between the two groups when the study populations are small, as in previous clinical studies. Our results suggest that EGD may be needed to detect UGI cancer in FIT+/CRC− patients. In addition, small intestine evaluation such as capsule endoscopy or abdominal computed tomography (CT) may be needed to consider for FIT+/CRC− patients without EGD or colonoscopy examination abnormalities who are suspected of having small intestine lesions. However, these suggestions are very cautious because the absolute PPV of FIT for these cancers in FIT+/CRC− patients were very low (0.08%, 1.07%, and 0.02% for esophageal, stomach, and small intestine cancers within 3 years after FIT). Given the numbers needed to scope to detect one case of UGI cancer, our results may not justify routine EGD investigation for all FIT+/CRC− patients. In addition to FIT results, the use of other clinical factors for predicting UGI cancer risk may allow the selection of subjects for UGI evaluation. GI symptoms, anemia, or family history of UGI cancer may be candidates for predicting high risks of UGI cancer [18,19,20]. Additionally, fecal hemoglobin (f-Hb) concentrations may need to be considered. Based on previous studies showing an association between higher f-Hb concentrations and increased CRC risk [21,22], high f-Hb concentration may also increase the yield of UGI evaluations among FIT+/CRC− patients. Further research is required to identify the risk factors for UGI cancer in FIT+/CRC− patients.

In the present study, the risk of hepatopancreatobiliary cancers was also higher in FIT+/CRC− patients than that in FIT− patients. Hepatocellular cancer usually progresses from liver cirrhosis, and some patients with biliary tract cancer also often have poor liver function. In patients with hepatobiliary cancer, deficiency of blood clotting factors and a decreased number of platelets due to deterioration of liver function may have resulted in positive FIT results. In patients with pancreatic cancer, bleeding from duodenal invasion of the pancreatic head cancer may have resulted in positive FIT results. Additionally, these results suggest that FIT positivity may be associated with diseases other than CRC. A recent study from Scotland reported that positive FOBT results were significantly associated with increased risks of non-CRC mortality (increased risk of death from respiratory, circulatory, and digestive diseases), suggesting that FOBT positivity could be used to alert CRC-screened participants to the risk of diseases other than CRC, regardless of the presence or absence of CRC [23].

We observed no significant differences in the risk of oral or throat cancer between FIT− and FIT+/CRC− patients. This may be because the oral cavity and throat are located at the top of the GI tract. Given that hemoglobin can be degraded or diluted as it moves through the GI tract, we expect that the lower the GI bleeding site, the higher the FIT positivity rate. In the present study, the adjusted ORs for oral or throat, esophageal, stomach, and small intestine cancer within 1 year after FIT were 1.01, 1.36, 1.49, and 1.74, respectively. The higher OR increases in cancers of the lower GI tract may support our expectation.

Another noteworthy finding of our study was that FIT+/CRC+ patients had higher risks of small intestine cancer and hepatopancreatobiliary cancer than FIT+/CRC− patients. CRC patients undergo abdominal CT to evaluate CRC stages, which may have resulted in increased detection of cancers in other abdominal organs. Additionally, patients with CRC may have a higher risk of developing synchronous cancer in other abdominal organs.

To our knowledge, this is the first large-scale population-based study to compare the risk of proximal cancers among FIT−, FIT+/CRC−, and FIT+/CRC+ patients in a nationwide CRC screening program with long-term follow-up. Nevertheless, the current study has several limitations. First, benign upper GI diseases, such as gastric or duodenal ulcers and severe esophagitis or gastritis, that could affect FIT results were not considered because it was difficult to accurately capture these benign diseases from diagnostic codes. Second, Groups 2 and 3 were defined using the diagnostic code for CRC rather than colonoscopic findings; thus, advanced adenoma was not considered. Some FIT+/CRC− patients may have had advanced adenoma, which may have affected positive FIT results. However, we considered patients with carcinoma in situ of the colon and rectum to be CRC+. Third, although most of the participants included in our study were expected to be asymptomatic because our study was based on a CRC screening program, we did not assess their symptoms. Finally, the stage of esophageal or gastric cancer was not considered because of the limitations of the NHIS–NHID.

In conclusion, FIT+/CRC− patients had significantly higher risks of esophageal, stomach, and small intestine cancers than FIT− patients. Our results indicate that positive FIT findings are associated with an increased risk of UGI cancer. However, because the absolute incidence of UGI cancer among FIT+/CRC− patients was low, other clinical risk factors should be considered when deciding whether to perform UGI assessment for FIT+/CRC− patients.

## Figures and Tables

**Figure 1 jcm-09-02172-f001:**
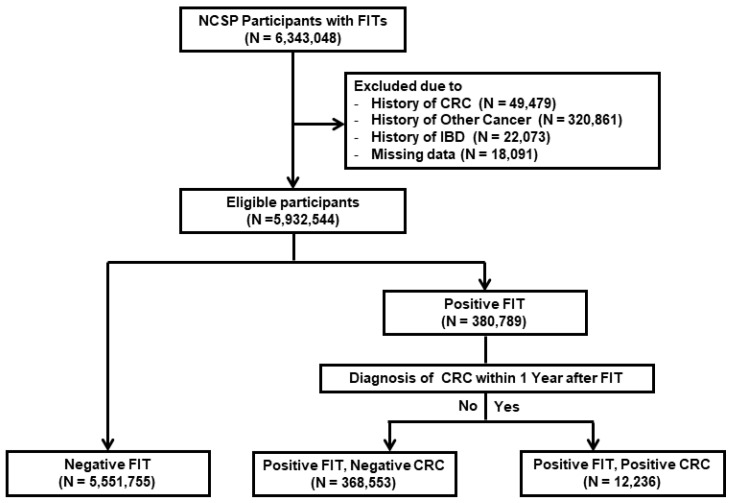
Flow chart of study participants. FIT, fecal immunochemical test; CRC, colorectal cancer; IBD, inflammatory bowel disease.

**Table 1 jcm-09-02172-t001:** Baseline characteristics of the study population according to FIT results and the presence of CRC.

Characteristics	Total *n* = 5,932,544	Group 1 (FIT−) *n* = 5,551,755	Group 2 (FIT+/CRC−) *n* = 368,553	Group 3 (FIT+/CRC+) *n* = 12,236	*p*-Value
Sex					
Male	2,609,985 (44.0)	2,411,853 (43.4)	190,178 (51.6)	7,954 (65.0)	<0.001
Female	3,322,559 (56.0)	3,139,902 (56.6)	178,375 (48.4)	4,282 (35.0)	
Age (years)	60.6 ± 8.2	60.6 ± 8.2	61.3 ± 8.6	64.4 ± 8.4	<0.001
50–59	2,938,401 (49.5)	2,761,549 (49.7)	173,165 (47.0)	3687 (30.1)	<0.001
60–69	1,976,063 (33.3)	1,849,210 (33.3)	122,075 (33.1)	4778 (39.1)	
70–79	900,122 (15.2)	833,366 (15.0)	63,459 (17.2)	3297 (27.0)	
≥80	117,958 (2.0)	107,630 (1.9)	9854 (2.7)	474 (3.9)	
Current smoking *	852,765 (14.4)	785,090 (14.1)	65,226 (17.7)	2449 (20.0)	<0.001
Alcohol intake **	1,968,636 (33.2)	1,828,320 (32.9)	135,258 (36.7)	5058 (41.3)	<0.001
Obesity (BMI ≥ 25 kg/m2) ***	2,091,385 (35.3)	1,955,118 (35.2)	131,968 (35.8)	4299 (35.1)	<0.001
Diabetic mellitus	1,428,926 (24.1)	1,329,211 (23.9)	96,229 (26.1)	3486 (28.5)	<0.001
Family history of any cancer ****	1,372,902 (23.1)	1,290,059 (23.2)	80,051 (21.7)	2792 (22.8)	<0.001
Use of aspirin	928,376 (15.6)	858,582 (15.5)	67,388 (18.3)	2406 (19.7)	<0.001
EGD within 1 year after FIT	708,624 (11.9)	655,388 (11.8)	48,773 (13.2)	4463 (36.5)	<0.001

Values are presented as mean ± SD or number (%). FIT, fecal immunochemical test; CRC, colorectal cancer; BMI, body mass index; EGD, esophagogastroduodenoscopy; FIT, fecal immunochemical test. * Missing data in 371,038 patients (6.3%). ** Missing data in 372,171 patients (6.3%). *** Missing data in 353,632 patients (6.0%). **** Missing data in 781,397 patients (13.2%).

**Table 2 jcm-09-02172-t002:** Locations and incidences of proximal cancers according to FIT results and the presence of CRC.

Cancer (ICD-10)	Group 1 (FIT−) *n* = 5,551,755	Group 2 (FIT+/CRC−) *n* = 368,553	Group 3 (FIT+/CRC+) *n* = 12,236
**Within 1 year after FIT**			
Overall proximal cancers (C00–17)	21,092 (0.38)	2516 (0.68)	277 (2.26)
Oral or throat cancer (C00–14) *	786 (0.01)	62 (0.02)	12 (0.10)
Esophagus cancer (C15)	1086 (0.02)	136 (0.04)	10 (0.08)
Stomach cancer (C16)	19,199 (0.35)	2308 (0.63)	233 (1.90)
Small intestine cancer (C17)	279 (0.005)	38 (0.01)	29 (0.24)
Hepatopancreatobiliary cancer (C22–25) **	9187 (0.17)	863 (0.23)	258 (2.11)
Other digestive organ cancer (C26)	116 (0.002)	16 (0.004)	22 (0.18)
**Within 2 years after FIT**			
Overall proximal cancers (C00–17)	31,486 (0.57)	3,430 (0.93)	335 (2.74)
Oral or throat cancer (C00–14) *	1681 (0.03)	134 (0.04)	13 (0.11)
Esophagus cancer (C15)	1838 (0.03)	213 (0.06)	18 (0.15)
Stomach cancer (C16)	27,898 (0.50)	3072 (0.83)	282 (2.30)
Small intestine cancer (C17)	541 (0.01)	59 (0.02)	33 (0.27)
Hepatopancreatobiliary cancer (C22–25) **	16,806 (0.30)	1521 (0.41)	350 (2.86)
Other digestive organ cancer (C26)	243 (0.004)	29 (0.01)	31 (0.25)
**Within 3 years after FIT**			
Overall proximal cancers (C00–17)	43,623 (0.79)	4467 (1.21)	386 (3.15)
Oral or throat cancer (C00–14) *	2559 (0.05)	208 (0.06)	18 (0.15)
Esophagus cancer (C15)	2629 (0.05)	293 (0.08)	24 (0.20)
Stomach cancer (C16)	38,322 (0.69)	3954 (1.07)	320 (2.62)
Small intestine cancer (C17)	831 (0.01)	87 (0.02)	38 (0.31)
Hepatopancreatobiliary cancer (C22–25) **	24,719 (0.45)	2193 (0.60)	433 (3.54)
Other digestive organ cancer (C26)	375 (0.01)	39 (0.01)	36 (0.29)

Values are presented as number (%). * Oral or throat locations included the lip, tongue, gum, mouth, palate, major salivary glands, parotid gland, oropharynx, nasopharynx, tonsil, piriform sinus, and hypopharynx. ** Hepatopancreatobiliary locations included the liver, gallbladder, biliary tract, and pancreas. FIT, fecal immunochemical test; CRC, colorectal cancer; ICD-10, International Classification of Diseases 10th revision.

**Table 3 jcm-09-02172-t003:** Risks of proximal cancers according to FIT results and the presence of CRC.

	Within 1 Year After FIT				Within 2 Years After FIT				Within 3 Years After FIT			
	Crude OR (95% CI)	*p*-Value	Adjusted OR ***	*p*-Value	Crude OR (95% CI)	*p-* Value	Adjusted OR ***	*p*-Value	Crude OR (95% CI)	*p*-Value	Adjusted OR ***	*p*-Value
Overall proximal cancers												
Group 1 (FIT−)	1 (ref)		1 (ref)		1 (ref)		1 (ref)		1 (ref)		1 (ref)	
Group 2 (FIT+/CRC−)	1.80 (1.73–1.88)	<0.001	1.47 (1.41–1.54)	<0.001	1.65 (1.59–1.71)	<0.001	1.38 (1.33–1.43)	<0.001	1.55 (1.50–1.60)	<0.001	1.31 (1.27–1.36)	<0.001
Group 3 (FIT+/CRC+)	6.07 (5.39–6.85)	<0.001	1.50 (1.32–1.69)	<0.001	4.94 (4.43–5.51)	<0.001	1.51 (1.35–1.69)	<0.001	4.11 (3.72–4.55)	<0.001	1.48 (1.33–1.64)	<0.001
Oral or throat cancer *												
Group 1 (FIT−)	1 (ref)		1 (ref)		1 (ref)		1 (ref)		1 (ref)		1 (ref)	
Group 2 (FIT+/CRC−)	1.19 (0.92–1.54)	0.191	1.01 (0.78–1.31)	0.934	1.20 (1.01–1.43)	0.042	1.03 (0.86–1.23)	0.749	1.23 (1.06–1.41)	0.005	1.06 (0.92–1.22)	0.424
Group 3 (FIT+/CRC+)	6.94 (3.92–12.27)	<0.001	3.01 (1.70–5.34)	<0.001	3.51 (2.03–6.06)	<0.001	1.81 (1.05–3.13)	0.034	3.20 (2.01–5.09)	<0.001	1.80 (1.13–2.86)	0.013
Esophagus cancer												
Group 1 (FIT−)	1 (ref)		1 (ref)		1 (ref)		1 (ref)		1 (ref)		1 (ref)	
Group 2 (FIT+/CRC−)	1.89 (1.58–2.26)	<0.001	1.36 (1.14–1.63)	<0.001	1.75 (1.52–2.01)	<0.001	1.32 (1.14–1.52)	<0.001	1.68 (1.49–1.90)	<0.001	1.29 (1.14–1.46)	<0.001
Group 3 (FIT+/CRC+)	4.18 (2.24–7.80)	<0.001	0.79 (0.42–1.47)	0.451	4.45 (2.80–7.08)	<0.001	1.11 (0.70–1.77)	0.656	4.15 (2.77–6.20)	<0.001	1.22 (0.82–1.83)	0.328
Stomach cancer												
Group 1 (FIT−)	1 (ref)		1 (ref)		1 (ref)		1 (ref)		1 (ref)		1 (ref)	
Group 2 (FIT+/CRC−)	1.82 (1.74–1.90)	<0.001	1.49 (1.42–1.55)	<0.001	1.66 (1.60–1.73)	<0.001	1.39 (.1.34–1.45)	<0.001	1.56 (1.51–1.61)	<0.001	1.33 (1.28–1.37)	<0.001
Group 3 (FIT+/CRC+)	5.59 (4.91–6.37)	<0.001	1.36 (1.19–1.55)	<0.001	4.67 (4.15–5.26)	<0.001	1.40 (1.24–1.58)	<0.001	3.86 (3.46–4.32)	<0.001	1.36 (1.21–1.52)	<0.001
Small intestine cancer												
Group 1 (FIT−)	1 (ref)		1 (ref)		1 (ref)		1 (ref)		1 (ref)		1 (ref)	
Group 2 (FIT+/CRC−)	2.05 (1.46–2.88)	<0.001	1.74 (1.24–2.45)	0.001	1.64 (1.26–2.15)	<0.001	1.45 (1.11–1.90)	0.007	1.58 (1.27–1.97)	<0.001	1.42 (1.13–1.77)	0.002
Group 3 (FIT+/CRC+)	47.28 (32.25–69.33)	<0.001	14.39 (9.75–21.25)	<0.001	27.75 (19.52–39.45)	<0.001	10.94 (7.65–15.65)	<0.001	20.81 (15.03–28.82)	<0.001	9.74 (7.00–13.55)	<0.001
Hepatopancreatobiliary cancer **												
Group 1 (FIT−)	1 (ref)		1 (ref)		1 (ref)		1 (ref)		1 (ref)		1 (ref)	
Group 2 (FIT+/CRC−)	1.42 (1.32–1.52)	<0.001	1.20 (1.12–1.28)	<0.001	1.37 (1.30–1.44)	<0.001	1.18 (1.12–1.24)	<0.001	1.34 (1.28–1.40)	<0.001	1.16 (1.11–1.22)	<0.001
Group 3 (FIT+/CRC+)	13.00 (11.47–14.73)	<0.001	4.86 (4.27–5.52)	<0.001	9.70 (8.71–10.80)	<0.001	4.37 (3.92–4.88)	<0.001	8.20 (7.45–9.04)	<0.001	4.11 (3.72–4.53)	<0.001

* Oral or throat locations included the lip, tongue, gum, mouth, palate, major salivary glands, parotid gland, oropharynx, nasopharynx, tonsil, piriform sinus, and hypopharynx. ** Hepatopancreatobiliary locations included the liver, gallbladder, biliary tract, and pancreas. *** Adjusted by age, sex, current smoking, alcohol intake, obesity, diabetic mellitus, family history of any cancer, use of aspirin, and esophagogastroduodenoscopy within 1 year after FIT. FIT, fecal immunochemical test; CRC, colorectal cancer; OR, odds ratio; CI, confidence interval.

**Table 4 jcm-09-02172-t004:** Comparisons of the risks of proximal cancers between FIT+/CRC− patients and FIT+/CRC+ patients.

	Within 1 Year After FIT				Within 2 Years After FIT				Within 3 Years After FIT			
	Crude OR (95% CI)	*p*-Value	Adjusted OR ***	*p*-Value	Crude OR (95% CI)	*p*-Value	Adjusted OR***	*p*-Value	Crude OR (95% CI)	*p*-Value	Adjusted OR ***	*p*-Value
Overall proximal cancers												
Group 2 (FIT+/CRC−)	1 (ref)		1 (ref)		1 (ref)		1 (ref)		1 (ref)		1 (ref)	
Group 3 (FIT+/CRC+)	3.37 (2.97s–3.82)	<0.001	1.02 (0.89–1.15)	0.802	3.00 (2.68–3.36)	<0.001	1.10 (0.98–1.24)	0.109	2.66 (2.39–2.95)	<0.001	1.13 (1.01–1.26)	0.031
Oral or throat cancer *												
Group 2 (FIT+/CRC−)	1 (ref)		1 (ref)		1 (ref)		1 (ref)		1 (ref)		1 (ref)	
Group 3 (FIT+/CRC+)	5.84 (3.15–10.83)	<0.001	2.98 (1.60–5.54)	<0.001	2.92 (1.65–5.17)	<0.001	1.76 (0.99–3.11)	0.053	2.61 (1.61–4.23)	<0.001	1.70 (1.05–2.75)	0.032
Esophagus cancer												
Group 2 (FIT+/CRC−)	1 (ref)		1 (ref)		1 (ref)		1 (ref)		1 (ref)		1 (ref)	
Group 3 (FIT+/CRC+)	2.22 (1.17–4.21)	0.015	0.58 (0.30–1.10)	0.094	2.55 (1.58–4.13)	<0.001	0.84 (0.52–1.37)	0.492	2.47 (1.63–3.75)	<0.001	0.95 (0.62–1.44)	0.804
Stomach cancer												
Group 2 (FIT+/CRC−)	1 (ref)		1 (ref)		1 (ref)		1 (ref)		1 (ref)		1 (ref)	
Group 3 (FIT+/CRC+)	3.08 (2.69–3.53)	<0.001	0.92 (0.80–1.05)	0.212	2.81 (2.48–3.18)	<0.001	1.01 (0.89–1.14)	0.940	2.48 (2.21–2.78)	<0.001	1.02 (0.91–1.15)	0.692
Small intestine cancer												
Group 2 (FIT+/CRC−)	1 (ref)		1 (ref)		1 (ref)		1 (ref)		1 (ref)		1 (ref)	
Group 3 (FIT+/CRC+)	23.05 (14.21–37.38)	<0.001	8.26 (5.07–13.44)	<0.001	16.89 (11.03–25.87)	<0.001	7.54 (4.91–11.59)	<0.001	13.20 (9.01–19.33)	<0.001	6.88 (4.69–10.11)	<0.001
Hepatopancreatobiliary cancer **												
Group 2 (FIT+/CRC−)	1 (ref)		1 (ref)		1 (ref)		1 (ref)		1 (ref)		1 (ref)	
Group 3 (FIT+/CRC+)	9.18 (7.98–10.56)	<0.001	4.06 (3.52–4.68)	<0.001	7.11 (6.32–8.00)	<0.001	3.72 (3.30–4.19)	<0.001	6.13 (5.52–6.81)	<0.001	3.53 (3.17–3.92)	<0.001

* Oral or throat locations included the lip, tongue, gum, mouth, palate, major salivary glands, parotid gland, oropharynx, nasopharynx, tonsil, piriform sinus, and hypopharynx. ** Hepatopancreatobiliary locations include the liver, gallbladder, biliary tract, and pancreas. *** Adjusted by age, sex, current smoking, alcohol intake, obesity, diabetic mellitus, family history of any cancer, use of aspirin, and esophagogastroduodenoscopy within 1 year after FIT. FIT, fecal immunochemical test; CRC, colorectal cancer; OR, odds ratio; CI, confidence interval.

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
