# Peer review of "Positive Fecal Immunochemical Test Results Are Associated with Increased Risks of Esophageal, Stomach, and Small Intestine Cancers"

_jcm, 2020, doi:10.3390/jcm9072172_

Round 1
Reviewer 1 Report
In the present study, the authors investigated the risk of proximal cancers after negative or positive FIT results in the Korean National Cancer Screening Program (NCSP). The risk of proximal cancers was compared among three groups: FIT− (n=5,551,755), FIT+/CRC− (n=368,553), and FIT+/CRC+ (n=12,236). The age- and sex-adjusted risks of esophageal, stomach, and small intestine cancers as well as overall proximal cancers within 1, 2, and 3 years after FIT were examined. The risk of esophageal, stomach, and small intestine cancers as well as overall proximal cancers were higher in FIT+/CRC− patients than those in FIT− patients. They concluded that FIT+/CRC− patients were at higher risk for esophageal, stomach, and small intestine cancers than were FIT− patients, and that positive FIT results were associated with these cancers. The data in this study has some value, but some of the results and discussions are not convincing. There seem to be several issues to be reconsidered as follows.
Major comments)
- As the participants were classified based on first FIT results and CRC diagnosed within 1 year, the risk of proximal cancers should be compared among three groups only within 1 year. What does the difference between a diagnostic risk within 1 year and within 3 years mean? If the participants were given esophagogastroduodenoscopy every year, the difference might mean frequency of diagnosis that was not associated with FIT.
- As the authors described in their discussion, they did not consider clinical factors that could affect cancer development, such as smoking, drinking, BMI, diabetes, family history, use of statins and aspirin, etc. The most important bias was whether or not the participants were given an upper gastrointestinal endoscopy within 1 year. The authors should analyzed these factors simultaneously by multivariate analysis.
- In the present study, the risk of hepatopancreatobiliary cancers was also higher in FIT+/CRC− patients than that in FIT− patients. The authors discussed that these results might be because hemobilia caused by hepatocellular carcinoma. This would generally be unacceptable. There would be some other bias that has affected the analysis.
- The authors also should exmine the frequency of lung cancer for which a FIT has no effect for comparison. It would prove the specificity of FIT.
Author Response
Reviewer 1.
In the present study, the authors investigated the risk of proximal cancers after negative or positive FIT results in the Korean National Cancer Screening Program (NCSP). The risk of proximal cancers was compared among three groups: FIT− (n=5,551,755), FIT+/CRC− (n=368,553), and FIT+/CRC+ (n=12,236). The age- and sex-adjusted risks of esophageal, stomach, and small intestine cancers as well as overall proximal cancers within 1, 2, and 3 years after FIT were examined. The risk of esophageal, stomach, and small intestine cancers as well as overall proximal cancers were higher in FIT+/CRC− patients than those in FIT− patients. They concluded that FIT+/CRC− patients were at higher risk for esophageal, stomach, and small intestine cancers than were FIT− patients, and that positive FIT results were associated with these cancers. The data in this study has some value, but some of the results and discussions are not convincing. There seem to be several issues to be reconsidered as follows.
Major comments)
As the participants were classified based on first FIT results and CRC diagnosed within 1 year, the risk of proximal cancers should be compared among three groups only within 1 year. What does the difference between a diagnostic risk within 1 year and within 3 years mean? If the participants were given esophagogastroduodenoscopy every year, the difference might mean frequency of diagnosis that was not associated with FIT.
Reply: Thank you for this comment, we agree with your concern. We sought to confirm the long-term data as well as the short-term data of FIT+/CRC- patients. Moreover, previous studies on topic similar to ours analyzed the risk of UGI cancer within 1, 2, or 3 years after FIT (Clin Gastroenterol Hepatol 2018, 16, 1237-1243: 3 years // Dig Liver Dis 2007, 39, 321-326: 1 year, 1-3 years, and ≥ 3 years // Scand J Gastroenterol 2002, 37, 95-98: 2 years). We described the results of previous studies in detail in the Discussion section. To compare our results with the results of previous studies, we analyzed the risk of proximal cancer within 2 and 3 years as well as within 1 year, based on these previous studies.
As the authors described in their discussion, they did not consider clinical factors that could affect cancer development, such as smoking, drinking, BMI, diabetes, family history, use of statins and aspirin, etc. The most important bias was whether or not the participants were given an upper gastrointestinal endoscopy within 1 year. The authors should analyzed these factors simultaneously by multivariate analysis.
Reply: Thank you for this comment, we agree with your concern. As the reviewer recommended, We adjusted potentially confounding variables, including current smoking, alcohol intake, obesity, diabetic mellitus, family history of any cancer, use of aspirin, and esophagogastroduodenoscopy (EGD) within 1 year after FIT as well as age and sex. We added these variables in the Table 1 and presented multivariable-adjusted ORs instead of age, sex-adjusted ORs in the Tables 3 and 4. Also, we revised the manuscript overall to reflect these results. The multivariable-adjusted ORs were similar to the previous results (age, sex-adjusted ORs).
In the present study, the risk of hepatopancreatobiliary cancers was also higher in FIT+/CRC− patients than that in FIT− patients. The authors discussed that these results might be because hemobilia caused by hepatocellular carcinoma. This would generally be unacceptable. There would be some other bias that has affected the analysis.
The authors also should exmine the frequency of lung cancer for which a FIT has no effect for comparison. It would prove the specificity of FIT.
Reply: Thank you for your comments. We deleted the sentence about hemobilia and added the following sentences in the Discussion section: “Hepatocellular cancer usually progresses from liver cirrhosis, and some patients with biliary tract cancer also often have poor liver function. In patients with hepatobiliary cancer, deficiency of blood clotting factors and a decreased number of platelets due to deterioration of liver function may have resulted in positive FIT results. In patients with pancreatic cancer, bleeding from duodenal invasion of the pancreatic head cancer may have resulted in positive FIT results.”
As the reviewer recommended, we analyzed the risk of lung cancer within 1 year after FIT.
|
|
Multivariable-adjusted OR |
P value |
Multivariable-adjusted OR |
P value |
|
Lung cancer |
|
|
|
|
|
Group 1 (FIT−) |
1 (ref) |
|
|
|
|
Group 2 (FIT+/CRC−) |
1.04 (0.96–1.13) |
0.334 |
1 (ref) |
|
|
Group 3 (FIT+/CRC+) |
3.57 (2.91–4.37) |
<0.001 |
3.42 (2.75–4.26) |
<0.001 |
As shown in the table above, the risk of lung cancer did not differ between FIT− and FIT+/CRC− patients. This result suggests that FIT may be specific for GI tract cancer. Because our study aimed to assess the risk of proximal GI cancers after positive FIT results, we did not present the risk of lung cancer.

Reviewer 2 Report
Title: Positive fecal immunochemical test results are associated with increased risks of esophageal, stomach, and small intestine cancers.
The authors aimed to assess the risk of proximal cancers after positive fecal immunochemical test. The topic of this work is interesting and the manuscript has been well written.
Author Response
Reviewer 2.
The authors aimed to assess the risk of proximal cancers after positive fecal immunochemical test. The topic of this work is interesting and the manuscript has been well written.
Reply: We appreciate your compliment.

Reviewer 3 Report
This study is very interesting, since it is a population-based study including more than 6 million patients, carried out at the National level in Korea, which analyzes the value of performing a gastroscopy in patients with a positive fecal occult blood test and negative colonoscopy.
It is the first study carried out at this scale, the methodology is correctly described, all ethical principles are followed, and the results are well described.
The authors highlight the fact that the diagnostic test for cancer in these patients was barium transit, supposing a limitation. In my opinion, a detail on the data of how many patients underwent gastroscopy and how many barium studies will enrich the results in this sense.
Finally, I see the great interest in this study, and I take the opportunity to congratulate the researchers for their work.
Sincerely,
Author Response
Reviewer 3.
This study is very interesting, since it is a population-based study including more than 6 million patients, carried out at the National level in Korea, which analyzes the value of performing a gastroscopy in patients with a positive fecal occult blood test and negative colonoscopy.
It is the first study carried out at this scale, the methodology is correctly described, all ethical principles are followed, and the results are well described.
The authors highlight the fact that the diagnostic test for cancer in these patients was barium transit, supposing a limitation. In my opinion, a detail on the data of how many patients underwent gastroscopy and how many barium studies will enrich the results in this sense.
Reply: Thank you for your comments. We added the number of patients who underwent esophagogastroduodenoscopy (EGD) within 1 year after FIT in the Table 1. Of 5,932,544 patients, 708,624 (11.9%) underwent EGD within 1 year after FIT. The proportion of those with EGD within 1 year after FIT was 11.8%, 13.2%, and 36.5% in groups 1(FIT-), 2 (FIT+/CRC-), and 3 (FIT+/CRC+), respectively (P<0.001). As recommended by the first reviewer, whether or not EGD was performed was adjusted in the analysis of proximal cancer risk (Table 3 & 4).
In Korea, the nationwide screening program for gastric cancer by EGD and upper gastrointestinal (UGI) series (barium meals) is performed as a part of the National Cancer Screening Program (NCSP) (Lee S, et al. Medicine 2015;94:e533). Screening by EGD was more strongly related to a diagnosis of localized gastric cancer than UGI series (Choi KS, et al. Br J Cancer 2015;112:608-612). In addition, if a patient is suspected of having malignancies in the UGI series, EGD is performed for biopsy in most cases. Therefore, we evaluated only EGD in our analyses.
Finally, I see the great interest in this study, and I take the opportunity to congratulate the researchers for their work.
Reply: Thank you for the invaluable comments. We appreciate the efforts made to review our manuscript.

Round 2
Reviewer 1 Report
The authors have revised thier manuscript according to the reviewer's comments. The problems pointed out have been improved.
Author Response
Thank you for your review.